# SARS-CoV-2 Spike Protein Amplifies the Immunogenicity of Healthy Renal Epithelium in the Presence of Renal Cell Carcinoma

**DOI:** 10.3390/cells13242038

**Published:** 2024-12-10

**Authors:** Maryna Somova, Stefan Simm, Jens Ehrhardt, Janosch Schoon, Martin Burchardt, Pedro Caetano Pinto

**Affiliations:** 1Department of Urology, University Medicine Greifswald, DZ7 J05.15, Fleischmannstraße 8, 17475 Greifswald, Germany; 2Institute of Bioinformatics, University Medicine Greifswald, Fleischmannstraße 8, 17475 Greifswald, Germany; 3Institute for Bioanalysis, Coburg University of Applied Sciences and Arts, Friedrich-Streib-Str. 2, 96450 Coburg, Germany; 4Department of Obstetrics and Gynecology, University Medicine Greifswald, Fleischmannstraße 8, 17475 Greifswald, Germany; 5Center for Orthopedics, Trauma Surgery and Rehabilitation Medicine, University Medicine Greifswald, Fleischmannstraße 8, 17475 Greifswald, Germany

**Keywords:** renal cell carcinoma, SARS-CoV-2, spike protein, renal proximal tubules

## Abstract

Renal cell carcinoma (RCC) is the most common form of kidney cancer, known for its immune evasion and resistance to chemotherapy. Evidence indicates that the SARS-CoV-2 virus may worsen outcomes for RCC patients, as well as patients with diminished renal function. Evidence suggests that the SARS-CoV-2 virus may exacerbate outcomes in RCC patients and those with impaired renal function. This study explored the unidirectional effects of RCC cells and the SARS-CoV-2 spike protein (S protein) on human renal proximal tubule epithelial cells (RPTECs) using a microphysiological approach. We co-cultured RCC cells (Caki-1) with RPTEC and exposed them to the SARS-CoV-2 S protein under dynamic 3D conditions. The impact on metabolic activity, gene expression, immune secretions, and S protein internalization was evaluated. The SARS-CoV-2 S protein was internalized by RPTEC but poorly interacted with RCC cells. RPTECs exposed to RCC cells and the S protein exhibited upregulated expression of genes involved in immunogenic pathways, particularly those related to antigen processing and presentation via the major histocompatibility complex I (MHCI). Additionally, increased TNF-α secretion suggested a pro-inflammatory response. Metabolic shifts toward glycolysis were observed in RCC co-culture, while the presence of the S protein led to minor changes. The presence of RCC cells amplified the immune-modulatory effects of the SARS-CoV-2 S protein on the renal epithelium, potentially exacerbating renal inflammation and fostering tumor-supportive conditions. These findings suggest that COVID-19 infections can impact renal function in the presence of kidney cancer.

## 1. Introduction

Renal cell carcinoma (RCC) is the most prevalent type of kidney cancer, often described as a “silent” disease due to its asymptomatic progression and resistance to conventional chemotherapy [1]. The intricate interplay within the tumor microenvironment (TME) is a critical factor in the RCC pathophysiology. This microenvironment consists of malignant cells interacting with non-malignant renal cells, immune cell populations, and the extracellular matrix. These interactions not only drive tumor growth but also modulate immune responses and therapeutic outcomes [2]. Investigating the dynamic relationship between RCC cells and the renal epithelium is essential for advancing targeted therapeutic strategies aimed at disrupting tumor progression [3].

In this study, we utilized a previously established microphysiological system (MPS) to co-culture RCC cells (Caki-1) with human renal proximal tubule epithelial cells (RPTECs) [4].

This in vitro model facilitated a comprehensive investigation of how RCC cells influence the functional dynamics of adjacent healthy renal cells, specifically focusing on changes in metabolic activity, transcriptional regulation, and cytokine secretion profiles.

This in vitro model enabled a thorough examination of how RCC cells affect the functional dynamics of neighboring healthy renal cells, with a particular emphasis on alterations in metabolic activity, transcriptional regulation, and cytokine secretion profiles.

The primary aim of this research is to assess whether RCC cells elicit alterations in the functional and molecular characteristics of non-malignant renal epithelial cells, potentially fostering a microenvironment conducive to tumor progression [5,6]. Furthermore, we aimed to investigate the impact of the SARS-CoV-2 S protein on these processes within the MPS framework, thus providing insights into the combined effects of RCC and the SARS-CoV-2 on renal cells.

Understanding the mechanisms underlying these cellular interactions is essential, as a key driver of RCC progression is its ability to evade immune surveillance [1].

The tumor itself is primarily recognized by the immune system through antigens presented by the major histocompatibility complex I (MHC-I) and detected by T cells [2]. However, RCC employs various mechanisms to escape immune detection, including the creation of a strongly immunosuppressive microenvironment that contributes to T-cell exhaustion [3]. In addition to immune evasion, we previously demonstrated that RCC cells can induce significant changes in neighboring non-malignant cells, promoting a pro-tumorigenic environment [4]. These changes include the upregulation of pro-inflammatory cytokines and metabolic reprogramming, such as a shift toward glycolysis [5], further highlighting the multifaceted ways RCC shapes its microenvironment. This complexity has become even more pronounced in the presence of additional systemic challenges, such as the SARS-CoV-2 viral infection.

Recent studies have emphasized the substantial impact of SARS-CoV-2 on the clinical outcomes of patients with RCC and other kidney conditions. The COVID-19 pandemic has posed significant challenges for oncology care, particularly among older individuals and those with compromised immune systems, such as cancer patients. Severe cases of COVID-19 have been associated with cytokine storms, lymphopenia, and multiorgan failure, all of which can exacerbate cancer prognosis. In particular, cancer patients face an increased risk of complications due to immunosuppressive therapies, comorbid conditions such as diabetes and hypertension, and the potential for COVID-19 to worsen due to certain cancer treatments [6]. Acute kidney injury (AKI) has emerged as a frequent complication in patients hospitalized with COVID-19, affecting over 20% of cases and often leading to poor clinical outcomes, including a 10% need for dialysis and heightened mortality rates. Although similar rates of AKI have been reported in other respiratory infections such as influenza, COVID-19 has been linked to more severe AKI and long-term complications, including chronic kidney disease (CKD) and increased mortality [7]. There are also growing concerns about the prolonged decline in kidney function among COVID-19 survivors, even in those who had not been hospitalized. Evidence has suggested that patients with COVID-19-associated AKI experience more significant long-term kidney function deterioration compared to AKI caused by other conditions. Furthermore, kidney damage associated with COVID-19 has been attributed to the presence of SARS-CoV-2 receptors, including the angiotensin-converting enzyme 2 (ACE2) receptor, in renal cells [8]. In general, healthcare disruptions during the pandemic led to delays in the diagnostics and treatment of malignancies like RCC, often resulting in more advanced-stage presentations and worse prognoses. The mechanism of SARS-CoV-2 infection, which involves the ACE2 receptor, presents added complexity for oncology patients. The mechanism of SARS-CoV-2 infection, which involves the ACE2 receptor, adds an extra layer of complexity for oncology patients.

The virus primarily enters cells through the ACE2 receptor and the TMPRSS2 protease [9,10]. The SARS-CoV-2 S protein virus binds to the ACE2 receptor, which is found in various organs, including the lungs, heart, intestines, and kidneys. TMPRSS2 then primes the S protein, allowing the viral membrane to fuse with the host cell membrane, enabling viral entry and replication [11].

Although the COVID-19 pandemic has been declared over, the long-term effects of COVID-19 still persist, affecting many individuals who have recovered from that acute illness. A deeper understanding of SARS-CoV-2’s effect on renal tissues and other organs is crucial for developing strategies to manage long-term COVID-19 and mitigate the effects of future outbreaks. Elucidating the interactions between SARS-CoV-2 and host cells, including its immune activation pathways and the resultant tissue damage, is critical for developing effective strategies to mitigate severe complications in both current infections and potential future pandemics. Considering the significant number of COVID-19 survivors at risk for long-term kidney complications, understanding the combined effects of SARS-CoV-2 and renal pathologies, such as RCC, is essential for optimizing post-COVID-19 care and informing future clinical management strategies.

In the current study, we aimed to characterize the bidirectional influences between RCC and RPTEC and how the SARS-CoV-2 S protein influences the immunogenicity of the renal epithelium in the presence of RCC, focusing on changes in gene expression, metabolic activity, and cytokine secretion. Utilizing our advanced in vitro RCC model, we demonstrated that the immunogenic effects of the SARS-CoV-2 S protein on renal epithelial cells are amplified in the presence of RCC, highlighting the synergistic impact of cancer and viral factors on renal cell behavior.

## 2. Materials and Methods

### 2.1. Cell Culture and MPS Preparation

Primary RPTECs (hRPTEC) and immortalized RPTECs (RPTEC-TERT1) as well as the RCC line Caki-1 were used in this study. RPTEC-TERT1 are non-malignant kidney epithelial cells that maintain most functional properties of primary renal proximal tubule cells, including drug transporter activity and responsiveness to nephrotoxic stimuli [12]. Caki-1 cells were derived from a metastatic site of clear cell renal carcinoma and represent an aggressive subtype of RCC, characterized by rapid proliferation, invasiveness, and resistance to apoptosis [13,14]. The cells were obtained from the American Type Culture Collection (ATCC). Cell culture supplements and consumables were acquired form Merck Lifesciences (Darmstadt, Germany), unless otherwise specified. All cell types were cultured in DMEM/F12 medium (Gibco, Thermo Fisher Scientific, Waltham, MA, USA) supplemented with 10% fetal bovine serum (FBS), 1% penicillin-streptomycin, 4 pg/mL Trio-do-L-thyronine, 10 µg/mL insulin-transferrin-selenium (ITS), 32 ng/mL hydrocortisone, and 10 ng/mL human recombinant epidermal growth factor (EGF), and maintained under the same conditions at 37 °C in a humidified atmosphere with 5% CO_2_. Medium was refreshed every 2 days, and cells were passaged at approximately 90% confluence [15]. The MPS model is based on the HUMIMIC Chip2 platform [16] (TissUse, Berlin, Germany). This system uses compressed air applied in a 4-set cycle to micro-pumps in the chip to control the perfusion (Figure 1). RPTEC cells were seeded into custom 3D-printed chambers designed to reconstruct renal tubules. The chambers were pre-filled with a low-density rat-tail collagen type I matrix (Corning Life Sciences, Corning, NY, USA) cross-linked using 30 mM Genipin and 1 M NaOH to enhance matrix stability, at a ratio of 1:20 (900 µL collagen + 50 µL Genipin + 50 µL NaOH). Cells were seeded at a density of 1 million cells per mL into the chambers, where they self-assembled into renal tubules. RCC spheroids were generated by embedding 100000 Caki-1 cells in an agar–collagen matrix (75 µL cell suspension + 25 µL collagen + 100 µL 2% Agar). The MPS platform was used to simulate physiological conditions and promote interactions between RPTEC tubules and RCC cells. The cell culture and MPS preparation were conducted following the procedures previously described [4,17]. The renal tubules mimic the hollow architecture of the kidney’s proximal tubules, while RCC aggregates display a compact and spherical architecture.

### 2.2. SARS-CoV-2 S Protein Incubation and ACE Staining

Renal tubules were incubated with SARS-CoV-2 S protein under different experimental conditions to evaluate the impact on cellular responses, as described in Figure 1K,L. Tubules in single culture or in co-culture with RCC were exposed to 1 ug/mL of recombinant SARS-CoV-2 S protein, derived from the Δ-variant (B.1.617.2), for a period of 5 days. Cells and supernatant were collected and stored at −80 °C for later analysis. Immunofluorescent staining for the localization of the ACE2 in RPTEC and RCC cells was performed in samples cultured without the S protein. To verify the internalization of S protein into RPTEC and RCC cells, fluorescently labelled S protein (S protein-488; 1 µg/mL)) was incubated with either renal tubules or RCC spheroids for 3h at 37 °C in cell culture media, in the presence or absence of the receptor-mediated endocytosis inhibitors Dyngo4a (50 µM; Abcam, Cambridge, UK) and PitStop (50 µM). Afterwards, samples were fixed with 2% paraformaldehyde (PFA), stained with 1:1000 Hoechst33342 (Stock solution 10 mM), and visualized using a BZ9000 fluorescent microscope (Keyence, Osaka, Japan). The assay materials are described in Appendix A.

### 2.3. Immune Secretions

The immune secretion profile of RPTECs and RCC exposed to the S protein was assessed by collecting supernatant from the MPS culture compartments after five days of perfusion. The cytokines and immune factors interleukin 1 (IL-1), interleukin 5 (IL-5), interleukin 1 (IL-6), interleukin 8 (IL-8), neutrophil gelatinase-associated lipocalin (NGAL), and the tumor necrosis factor alfa (TNF-α) were quantified using R&D Systems^®^ ELISA Kits (Minneapolis, MN, USA) following the manufacturer’s instructions, as previously described. Additionally, extracellular lactate and glucose levels were also measured using the Lactate-Glo™ assay. The description of the ELISA consumables can be found in Appendix B.

### 2.4. RNA Extraction, Sequencing and Bioinformatics Analysis

Total RNA was extracted from the renal tubule matrices using the RNeasy Mini Kit (Qiagen, Hilden, Germany) according to the manufacturer specification and as previously described. RNA-Sequencing was performed by NovoGene Europe (Cambridge, UK). Samples included three replicates per condition tested. Sequencing depth was between 12,032,453 and 14,734,138 read-pairs, and reads-mapping was performed using NextGenMap (with default parameters for very sensitive) on the GRCh38.p14 human genome assembly (Ensembl). Gene counting was performed using featureCounts (version 2.0.4) via the R package. The differentially expressed genes (DEGs) were identified using DESeq2 [18] with the standard parameters between CC, CC_SP, 3D and SP samples. The DEGs below an adjusted *p*-value of 0.05 were split based on their log2 foldchange (FC) in 2D upregulated (positive FC) and 2D downregulated (negative FC). The R packages complex Heatmap, enhanced Volcano, clusterProfiler, and org.Hs.eg.db were used to identify overrepresented pathways for GeneOntology (GO) “Biological Process” (BP), Kyoto Encyclopedia of Genes and Genomes (KEGG) and the creation of Volcano plots, interaction networks and heatmaps.

### 2.5. Metabolic Activity

The metabolic activity of the cells was evaluated using the Seahorse XF Analyzer (Agilent Technologies, Santa Clara, CA, USA). Oxygen consumption rates (OCRs) and extracellular acidification rates (ECARs) were measured to assess oxidative phosphorylation and glycolysis, respectively. The Cell Mito Stress Test Kit was employed to determine the basal metabolic rates and ATP production rates under each condition. Sample preparation and analysis were carried out as previously described and according to the manufacturer’s specifications.

### 2.6. Cell Viability

Cell viability was assessed using a combination of lactate dehydrogenase (LDH) leakage, caspase activity and extracellular ATP detection assays. Supernatant samples from each condition were tested using Glo^TM^ assays (Promega, Madison, WI, USA). The assay conditions are described in Appendix C.

### 2.7. Exploratory Statistics and Data Plotting

ELISA, luminescent and metabolic assay results were analyzed and plotted using GraphPad Prism 8 (La Jolla, CA, USA). Statistically significant differences were estimated using a 2-way ANOVA. All assays were performed using three biological replicates unless stated otherwise. Immunofluorescent images were processed using the open-source software Fiji (https://imagej.net/software/fiji, accessed on 29 October 2024).

## 3. Results

### 3.1. ACE Expression and the SARS-CoV-2 S Protein Internalization

In the renal tubules, ACE2 was prominently localized along the basolateral membrane, facing the exterior of the tubule. In the RCC spheroids, ACE2 distribution appeared more diffused, reflecting a less organized basolateral membrane (Figure 2A–C). Incubation of the renal tubules with fluorescently labelled S protein revealed the intracellular presence of the viral protein after a 3 h exposure. However, in the presence of specific receptor-mediated endocytosis (RME) inhibitors, no S protein was observed in the renal tubules. Dyngo4a and PitStop block the activity of Dynamin and the assembly of Clathrin vesicles, respectively (Figure 2D–F). These observations underlined the fact that SARS-CoV-2 enters RPTEC via an RME mechanism. On the other hand, in RCC spheroids, no fluorescent S protein was observed after incubation using the same conditions and in the presence or absence of RME inhibitors. This observation suggested that the SARS-CoV-2 S protein poorly interacted with RCC cells.

### 3.2. Gene Expression Changes Across Experimental Conditions

The common gene expression across three experimental conditions is illustrated with a Venn diagram (Figure 3A). The analyzed experimental conditions included RPTEC-TERT cells co-cultured with RCC (CC), RPTEC-TERT cells exposed to the SARS-CoV-2 S protein, and RPTEC-TERT cells exposed to both RCC cells and the SARS-CoV-2 S protein simultaneously (CC_SP). The CC condition induced alterations in the expression of 94 genes, whereas the S protein condition resulted in differential expression of 117 genes. The CC_SP condition exhibited the most extensive impact, with 126 genes displaying altered expression. Importantly, 25 genes were consistently differentially expressed across all three conditions. Additionally, 10 genes were shared between the CC and CC_SP conditions, 64 genes were commonly altered between the SP and CC_SP conditions, and 6 genes were affected by both CC and SP conditions. The DEGs in the pairwise comparisons (Figure 3B–D) showed the highest amount of DEGs between 3D and SP followed by 3D and CC_SP.

Gene Set Enrichment Analysis (GSEA, Figure 4A) revealed a pronounced upregulation of pathways involved in antigen processing and presentation via the major histocompatibility complex (MHC) Class I, when renal tubules were co-cultured with RCC and exposed to S protein (CC vs. CC_SP up). Associated pathways included the TAP-independent variant and the presentation of endogenous peptide antigens through the endoplasmic reticulum (ER). This suggested that the introduction of S protein enhanced the immune response in RPTEC. On the other hand, the downregulation after S protein exposure (CC vs.CC_SP down) was related to the chromosome segregation, nuclear division, and mitotic spindle organization. This observation indicated potential disruptions in normal processes of cell division triggered by S protein in the presence of RCC. In RPTECs exposed to the S protein only (3D vs. SP up), there was a notable upregulation of metabolic pathways, relating to cholesterol biosynthetic processes, indicating a slight metabolic shift in RPTECs. These alterations likely represented the cells’ adaptive response to viral-induced stress, aiming to maintain membrane integrity. Contrary to the condition where RCC was present, no upregulation of immunogenic pathways was observed after S protein exposure alone. The Cnet plots showed a high degree of interconnectivity of the DEGs from the overrepresented pathways (Figure 4B). To compare the changes in DEG between the conditions regarding T-cell activity and antigen presentation (Figure 4C), we performed hierarchical clustering. The expression pattern in the heatmaps for T-cell- and antigen-related DEGs and their expression showed that CC_SP and SP were more closely clustered together than CC and 3D.

### 3.3. Immune Modulation in RPTEC and RCC Co-Culture in Response to SARS-CoV-2 S Protein

In the renal tubules, the secretion of TNFα exhibited an upregulation upon exposure to the S protein, indicating the induction of a pro-inflammatory response triggered by the viral component. This inflammatory response was further amplified in the RCC co-culture condition, where TNF-α levels were significantly elevated in the presence of the S protein, relative to its absence (Figure 5A). The addition of S protein resulted in a slight elevation in NGAL levels in a single culture. In contrast, NGAL levels were significantly elevated in the RCC co-culture in the presence of the S protein. Interestingly, NGAL expression was substantially reduced in the RCC co-culture in the absence of the viral protein, suggesting a potential suppressive effect exerted by RCC cells on NGAL secretion by the renal tubules, without the challenge of viral components (Figure 5B). IL-8 expression followed a similar trend, showing a significant upregulation in the single culture upon exposure to the S protein. However, in the RCC co-culture condition, IL-8 levels remained unaltered despite the presence of the S protein (Figure 5C). The expression of IL-1 was stable across all experimental conditions, indicating that this cytokine may not be responsive to the stress signals generated by the viral component or the presence of RCC cells, suggesting a more conserved role in these contexts (Figure 5D). In contrast, IL-5 exhibited the most striking differential expression between the two culture conditions. In the single culture, IL-5 levels were nearly undetectable, indicating negligible production of this cytokine by RPTEC-TERT1 cells when cultured alone. However, in the RCC co-culture, IL-5 expression was significantly elevated, pointing to a strong induction of IL-5 production in the presence of RCC cells that was not affected by S protein (Figure 5E).

### 3.4. Metabolic Adaptations of RPTEC and RCC Co-Culture to SARS-CoV-2 S Protein

The metabolic profiling of the renal tubules and RCC spheroids was investigated using the Seahorse analyzer. In addition, glucose consumption and lactate production were also determined. In single culture, the renal tubules showed a balance between glycolysis and Ox/Phos metabolic activity, favoring mitochondrial respiration (Ox/Phos: 59.1%, Glyco: 40.9%). The addition of S protein slightly reversed this trend, with Ox/Phos representing 43% and glycolysis 57% of energy production. In co-culture with RCC, the metabolism of the renal tubules tipped substantially in favor of glycolytic activity, with 72% of the total energy production (Figure 6A,B). In the presence of S protein, this shift subsided, with glycolysis still the favored energetic pathway; however, this reduced to 55% of the total metabolic output. These results showed that RCC altered the activity of the renal tubule, while the presence of S protein yielded minor effects. Noteworthy is the fact that in co-culture with renal tubules, RCC enhanced the Ox/Phos activity. This shift reverted to a predominantly glycolytic state with the addition of S protein. No significant differences in glucose consumption were observed across the tested conditions, as the glucose concentration in the culture media remained unchanged after 5 days compared to the initial levels. Extracellular lactate was only slightly increased in the presence of S protein in the renal tubules without RCC co-culture (Figure 6C,D). The extracellular acidification rate (ECAR), oxygen consumption rate (OCR), proton efflux rate (PER), and profiles are described in Appendix D (Figure A1).

### 3.5. Effects of SARS-CoV-2 S Protein on Cell Viability

Lactate dehydrogenase (LDH) release, a well-established marker of cell membrane integrity and cytotoxicity, revealed that in single culture conditions, the presence of the S protein reduced LDH levels compared to conditions without the S protein in the renal tubules. In the RCC co-culture condition, LDH release remained unaltered, regardless of S protein presence. Extracellular caspase 3/7 activity, a marker of apoptosis, remained relatively constant across all experimental conditions, regardless of the S protein’s or RCC’s presence (Figure 7A–C). This suggested that neither the S protein nor the RCC co-culture significantly influences apoptotic processes in RPTEC cells under the tested conditions. Extracellular ATP levels were similarly stable across all conditions. This stability reiterated that the renal tubules maintained their membrane integrity in the presence of S protein and when co-cultured with RCC.

## 4. Discussion

This study provided insights into the interplay between RCC and the renal epithelium in the presence of the SARS-CoV-2 S protein, and it further validated our approach to employ an MPS with co-culture of RCC and renal proximal tubule epithelium cells [4], to investigate kidney cancer pathophysiology in vitro and its complex interactions. Moreover, this study also showed that the physiological effects of individual viral components on different tissues can be investigated in a controlled in vitro system. In the context of SARS-CoV-2, this fact is relevant given that the S protein is the target in mRNA-based vaccines developed for population-induced immunity against the virus [19].

SARS-CoV-2 S protein internalization is known to be facilitated by ACE2 via RME [9,20], a process corroborated by our results. Both renal tubules and RCC spheroids express ACE on their basolateral membrane, with the density of the receptor seemingly more pronounced in RTPEC, displaying an evident basal polarization. Interestingly, no internalization of S protein was observed in RCC cells. This may be due to differential expression levels of ACE2, which are reported to be higher in healthy kidney cells, and their presence in RCC may be downregulated or altered due to the tumor’s pathophysiology [21,22]. These findings suggest that RCC cells may be less prone to direct viral infection and that the potential effects of SARS-CoV-2 on renal activity were indirectly caused by the presence of RCC.

A particularly significant finding was the upregulation of genes related to antigen processing and presentation via MHC Class I pathways in RPTECs co-cultured with RCC and exposed to the SARS-CoV-2 S protein. This suggested that S protein enhanced the immunogenicity of RPTEC by increasing the expression of MHC Class I components, reflecting an adaptive response to viral elements, potentially alerting cytotoxic T cells to the presence of infected cells. This activation in the presence of the TME has complex implications. RCC created an immunosuppressive TME characterized by the deregulation of MHC Class I proteins, abundant cytokine secretion, and the recruitment of regulatory T cells. These adaptations enable tumors to inhibit the activity of cytotoxic T cells while maintaining a population of sequestered immune cells [23,24]. In tandem, we also observed a heightened inflammatory state in the presence of both the S protein and RCC, particularly the upregulated secretion of TNF-α, a critical mediator in the immune response, known for its role in promoting inflammation, apoptosis, and necrosis under various pathological conditions. Interestingly, the Caki-1 RCC spheroids did not express TNF-α [4], and the enhanced TNF-α production indicated that RCC cells exacerbate the immune activation initiated by the S protein in the non-tumor cells. Clues to how RCC spheroids manipulated the local immune environment were found on the expression of other inflammatory cytokines. The secretion of IL-1, IL-5 and IL-8 is not substantially impacted by the presence of S protein. These factors were involved in the recruitment of macrophages, eosinophils, and neutrophils, respectively [25,26,27]. Noteworthy is the fact that IL-8 was elevated in the renal tubules during RCC co-culture and that IL-5 is only secreted by RCC spheroids, indicating that the malignant cells were attempting to actively recruit cytokine-secreting immune cells.

In addition to the immune alterations, metabolic changes in healthy renal tubules were also observed when co-cultured with RCC cells. The shift toward glycolysis in the renal epithelium is a hallmark of cancer metabolism. While the metabolic shift was more pronounced in the presence of RCC, the introduction of SARS-CoV-2 S protein only marginally altered the energy production balance, slightly enhancing glycolysis. These findings suggested that while RCC induced a substantial metabolic shift in surrounding non-malignant cells, SARS-CoV-2 S protein might not have as profound an effect on cellular metabolism as on immune responses. Interestingly, while SARS-CoV-2 S protein only slightly altered the glycolytic shift, it did not appear to reverse the RCC-induced metabolic reprogramming [28]. SARS-CoV-2 has been implicated in causing metabolic disturbances in various tissues, including the kidneys. The conditions simulated in our study differ from an actual viral infection by SARS-CoV-2, and our results suggested that any impact of S protein alone on cellular metabolism is overshadowed by the metabolic reprogramming driven by RCC [29,30,31]. In our assays, we observed substantial variability in the metabolic activity measurements of the renal tubules. The data presented consist of two biological replicates per condition, and no significant statistical differences were observed. We hypothesize that these inconsistencies were derived from metabolic flexibility of the renal tubules that can reflect minor changes in independent experiments (Appendix D). On the other hand, RCC activity was consistently more stable (i.e., less intra-experiment variability), hence our assumption that the metabolic activity of the tumor spheroids was constant and less sensitive to the changes in the environment.

Analysis of the cellular effluent after the co-culture period (5 days) showed glucose consumption to be seemingly unaltered. Therefore, glucose levels in the culture media were well above the metabolic requirements of the RCC spheroids and renal tubules in dynamic culture, a fact that may influence metabolic activity by favoring glycolytic activity. Extracellular lactate levels were similarly unaltered across all conditions, and a slight increase in the presence of S protein in the renal tubules is likely a reflection of the metabolic shift observed. This observation correlated to a decrease in LDH release observed under the same experimental conditions. It is likely that while altering their metabolism, healthy renal tubules experience diminished LDH activity to limit lactate accumulation, preventing excessive acidification and further enzymatic deregulation [32]. This phenomenon was not observed in the presence of RCC, likely due to the ability of tumors to adapt to acidic conditions thanks to their metabolic resilience. The experimental conditions tested in our study did not impact cell viability, as seen by the stable levels of LDH release, extracellular caspase activity, and ATP contents.

Our findings indicated that changes in the microenvironment of renal tubules promoted by the presence of RCC exacerbated their immune response to the SARS-CoV-2 S protein. Immune secretions and metabolic reprograming may account for the effects observed, but likely further mechanisms involved in intracellular communication were involved, considering that the tubules and spheroids share the same recirculating fluidic (i.e., culture media) environment. The recruitment of macrophages or lymphocytes by either the healthy epithelium or the malignant cells was not investigated, since our MPS model did not incorporate immune cells; consequently, the upregulated immunogenic activity observed in the cells could not be functionally executed. Future iterations of the model can incorporate circulating immune cell populations (e.g., human peripheral blood mononuclear cells) to increase the physiological representation of the system [33]. Moreover, our system was based on the Caki-1 RCC cells, which could spontaneously develop consistent and stable spheroids, and represented a metastatic type of RCC (i.e., highly malignant). This fact benefited the modulation of the MPS environment by the tumor cells but limited our model to the effects of a particular RCC phenotype. Further models could incorporate other RCC types (e.g., primary tumor-derived cells) to evaluate the extent of the tumor-induced immunogenicity [34].

The results of this study shed light on the fact that RCC patients infected with SARS-CoV-2 may experience exacerbated renal inflammation and immune dysregulation, increasing their risk of AKI and other complications involving renal damage [6,35]. Further research into the long-term consequences of SARS-CoV-2 infection in cancer patients is required, focusing on how viral-induced immune and metabolic changes might influence tumor progression and patient survival. Characterizing the differential expression of ACE2 and other viral receptors in RCC could also help clarify the mechanisms behind the virus’s interactions with tumor tissues [36]. Clinical observations showed that cancer patients, particularly those with solid tumors like RCC, are at higher risk for severe outcomes from COVID-19 [37,38]. Severe cases of COVID-19 were often accompanied by cytokine storms, characterized by high levels of systemic inflammation [38]. For RCC patients, already compromised by the immunosuppressive nature of the tumor and its treatment, the additional inflammatory burden imposed by SARS-CoV-2 might lead to worsened clinical outcomes, compounded by impaired renal function.

## 5. Conclusions

In this study, we demonstrated that RCC cells amplify the immune-modulatory effects of the SARS-CoV-2 S protein on renal epithelial cells, particularly through increased inflammatory cytokine production and metabolic shifts. These findings suggested that RCC patients may be at higher risk for complications from COVID-19, including exacerbated renal inflammation and impaired immune responses. Future strategies should focus on mitigating these effects to improve clinical outcomes for RCC patients during viral infections.

## Figures and Tables

**Figure 1 cells-13-02038-f001:**
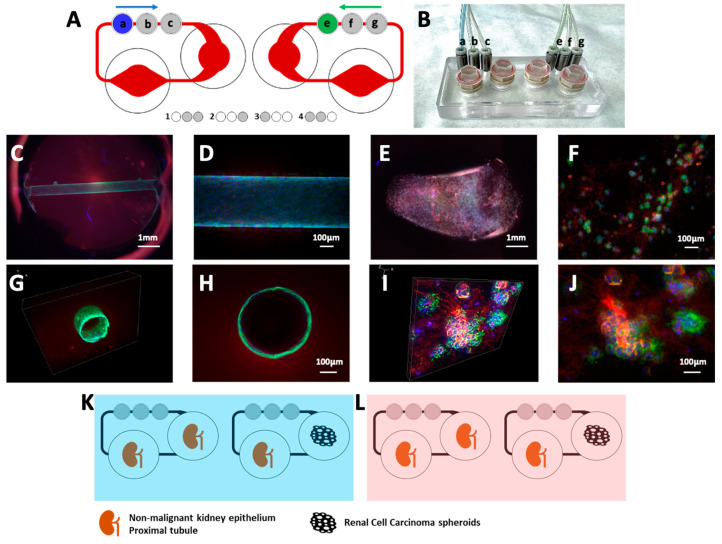
MPS experimental set-up and characterization of renal tubules and RCC spheroids. (**A**) Schematic representation of the Humimic Chip2 for co-culturing RPTEC tubules and RCC spheroids. The platform included two microfluidic circuits with two compartments each. Perfusion was driven by three micropumps in each circuit (e–g) activated by compressed air from an external unit, using a 4-step cycle. (**B**) Humimic chip2 with tubing and adaptors connected. (**C**) Non-malignant kidney epithelial cells forming tubular structures in the microfluidic channel under magnification 2×. (**D**) Tubular epithelium under magnification 10×. (**E**) RCC spheroids embedded in Agar-collagen gel (2×). (**F**) RCC spheroids detail at 10×. (**G**,**H**) Cross-sectional views of epithelial structures showing the organization of RPTEC within the tubules. (**I**,**J**) Cross-sectional views of RCC spheroid formed under flow conditions within the chip. (**K**) Experimental set-up of the renal tubules in single culture and in co-culture with RCC without SARS-CoV-2 S protein, highlighted in blue (**L**) Experimental set-up of the renal tubules in single culture and in co-culture with RCC under SARS-CoV-2 S protein influence, highlighted in soft pink.

**Figure 2 cells-13-02038-f002:**
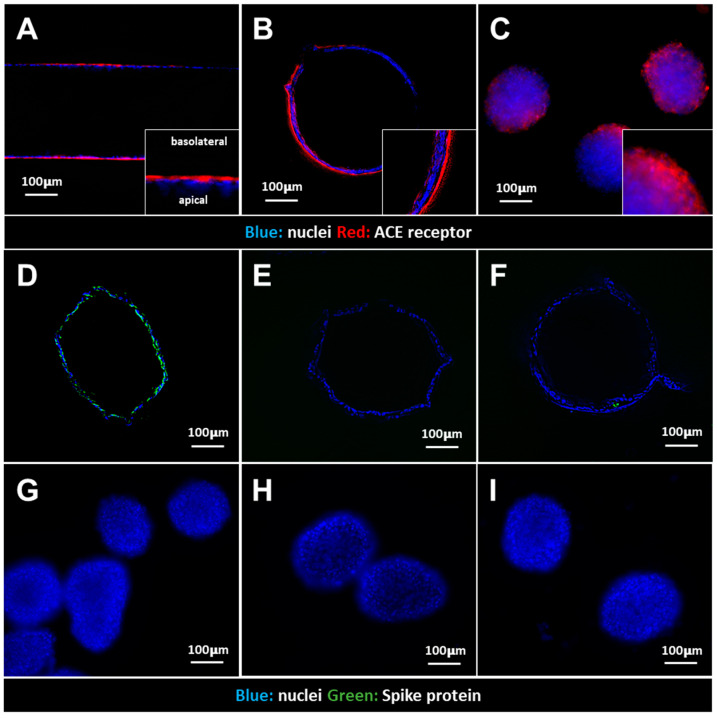
ACE2 expression and SARS-CoV-2 S protein uptake by renal tubules and RCC spheroids. (**A**,**B**) ACE expression localized on the basolateral membrane of the renal tubules. (**C**) Diffuse expression of ACE on the membrane of RCC spheroids. (**D**) Fluorescent SARS-CoV-2 S protein uptake by the renal tubules (cross-section). (**E**,**F**) Absence of SARS-CoV-2 S protein uptake after exposure with the addition of RME inhibitors Dyngo4a and PitStop. No S protein uptake was observed in RCC spheroids (**G**), or with the addition of RME inhibitors Dyngo4a and PitStop (**H**,**I**).

**Figure 3 cells-13-02038-f003:**
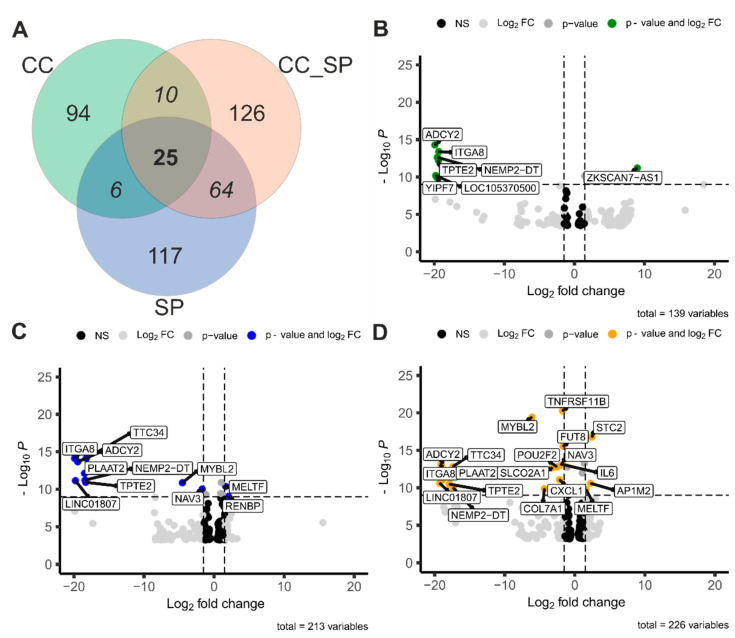
Differential gene expression analysis of renal tubules exposed to RCC and SARS-CoV-2 S protein. (**A**) Venn diagram illustrating the overlap of differentially expressed genes among three conditions: co-culture (CC), co-culture with SARS-CoV-2 S protein (CCSP), and S protein alone (SP). (**B**) Comparison between DEG in the CC_SP and SP conditions (**C**) Comparison between 3D (renal tubule) and CC_SP. (**D**) Comparison between 3D (renal tubule) and S protein. These volcano plots demonstrated distinct gene expression profiles under co-culture conditions and the effects of SARS-CoV-2 S protein, revealing important changes in immune response and cancer-related pathways.

**Figure 4 cells-13-02038-f004:**
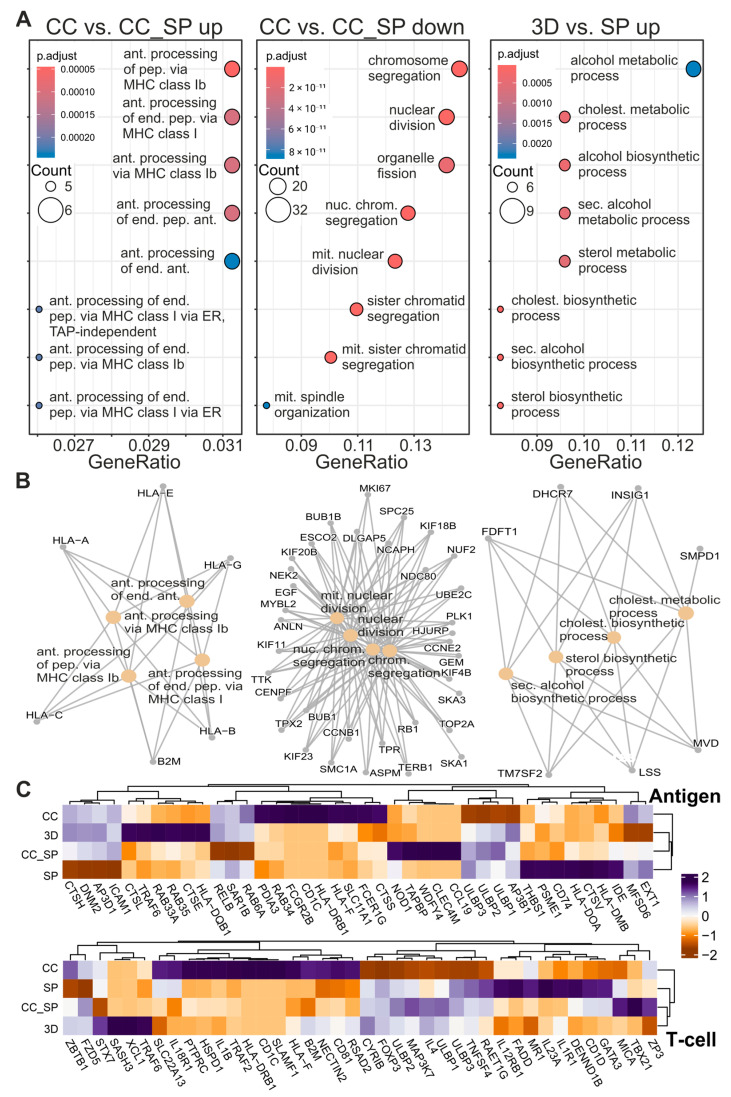
Pathway and gene enrichment analysis of renal tubules exposed to RCC and SARS-CoV-2 S protein. (**A**) Dot plots displaying enriched biological processes for genes upregulated (**left**) and downregulated (**middle**) in co-culture with RCC exposed SARS-CoV-2 S protein (CC_SP) versus co-culture alone (CC), and upregulated processes in the S protein treatment alone (**right**). Key pathways included antigen processing via MHC class I in the upregulated genes for CC_SP versus CC, and chromosome segregation in the downregulated genes. (**B**) Network of enriched biological processes and associated genes, connecting antigen processing, chromosome segregation, and metabolic processes such as cholesterol biosynthesis. Each node represented a biological process, illustrating the relationship between immune response, cell division, and metabolism under different experimental conditions. (**C**) Heatmaps of gene expression levels (z-score normalized) for antigen processing (**top**) and T cell activation markers (**bottom**) across experimental conditions (CC, CC_SP, SP, 3D). The color scale represented the expression levels, with purple indicating upregulation and yellow indicating downregulation. These heatmaps highlighted the differential expression of immune-related genes and pathways, showing how the SARS-CoV-2 S protein affects antigen presentation and T cell activation.

**Figure 5 cells-13-02038-f005:**
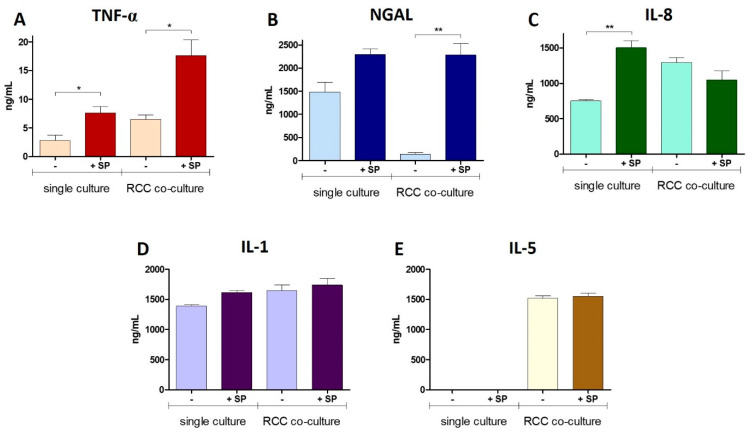
Cytokine secretion by the renal tubules under the influence of RCC and SARS-CoV-2 S protein. (**A**) TNF-α levels substantially increased in RCC co-cultures upon S protein exposure (**B**) NGAL secretion was significantly reduced in RCC co-culture without S protein exposure (**C**) IL-8 levels significantly increased upon S protein exposure in single culture but remained unchanged in RCC co-cultures (**D**) IL-1 expression levels showed no significant variation across all conditions, remaining stable in both single and RCC co-culture conditions (**E**) IL-5 was undetectable in single cultures, but its expression dramatically increased in RCC co-culture with our without S protein. Statistical significances were determined using a two-tailed unpaired *t*-test (* *p* < 0.05; ** *p* < 0.01).

**Figure 6 cells-13-02038-f006:**
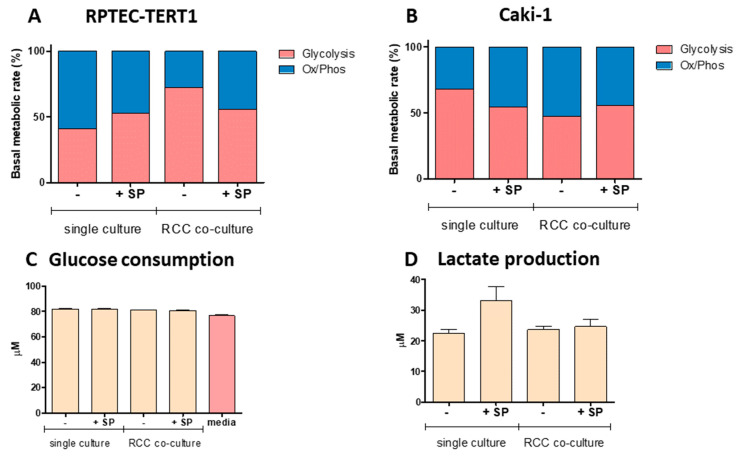
Metabolic activity of renal tubules and RCC spheroids under the influence of SARS-CoV-2 S protein. (**A**) Basal metabolic rate in the renal tubules, showing the proportion of energy production via glycolysis (red) and oxidative phosphorylation (OxPhos, blue). In both single culture and RCC co-culture, the introduction of S protein led to a slight increase in glycolysis while OxPhos remains relatively stable. (**B**) Basal metabolic rate in the RCC spheroids, displaying a similar energy production profile, across all conditions tested. Glycolysis predominates across all conditions, with S protein exposure causing minimal changes in the metabolic distribution. (**C**) Glucose consumption levels remained consistent across all conditions over the culture period, the initial glucose concentration in the culture media was shown in soft pink. (**D**) Lactate production increased slightly in single culture upon S protein exposure, while remaining stable in RCC co-culture.

**Figure 7 cells-13-02038-f007:**
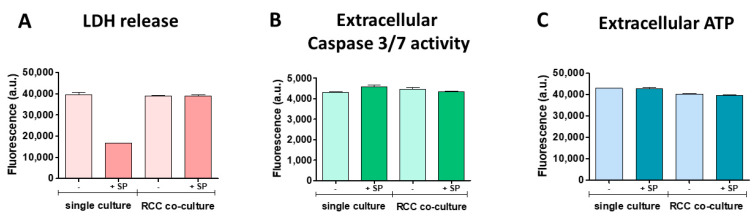
Renal tubules cytotoxicity under the influence of RCC and SARS-CoV-2 S protein. (**A**) LDH release levels. (**B**) Extracellular caspase 3/7 activity, (**C**) Extracellular ATP levels. Results show only a slight decrease in LDH release in the presence of S protein, highlighting that the conditions tested had no measurable effect on cell viability.

## Data Availability

Sequencing data and differential gene expression (DEG) analysis are available in the National Center for Biotechnology Information (NCBI) Gene Expression Omnibus (GEO) repository under accession number GSE244498, accessible with the token: qhcfocsqhhmjfgp.

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
