# Peer review of "SARS-CoV-2 Spike Protein Amplifies the Immunogenicity of Healthy Renal Epithelium in the Presence of Renal Cell Carcinoma"

_cells, 2024, doi:10.3390/cells13242038_

Round 1

Reviewer 1 Report

Comments and Suggestions for Authors

In their work, Dr. Somova and collaborators present a study on the combined effects of exposure to the Spike protein of SARS-CoV-2 (delta variant) in combination with the presence of metastatic renal cell carcinoma cells on the epithelial cells of the renal proximal tubule. The work investigates the effects at the metabolic, immunomodulatory and trascriptional levels.

Although the work exploits a very fine microphysiological system for the study of the renal epithelium, some analyzes are described in an unclear and somewhat too approximate manner.

I have some major concerns about the analyses presented:

1) In the presentation of the data regarding the expression of the ACE2 receptor and the internalization of the Spike protein, the different internalization capacity of the Spike protein between RPTEC and RCC cells is described, but in figure 2 I cannot find the data presented for the RCC spheroids. The lack of internalization of SP in RCC cell is also reported during discussion of the data, but is not shown in the Result section.

2) Why is gene set enrichment analysis not reported for 3D versus CC? The immunomodulatory effect of CC+SP is described, but what about exposure to CC alone?

Related to these analyses, the gene names in the heatmaps of Figure 4C are not clearly readable.

3) Why was the level of cytokine secretion not measured only in RCC (-/+ SP) cells? Perhaps this control could help define additional effects. It appears that RCC co-culture always has a greater effect than SARS-CoV2 SP exposure dependence.

4)In the metabolic adaptation analysis, in figure 6A-B, the metabolic rate are reported for RPTEC (A) and RCC (B), respectively. Is not clear why Caki-1 (RCC cells) are co-cultured with RCC. Which condition is reported in Figure 6B?. Please clarify better, because is not properly clear. Why tipical Seahorse profiles are not reported? Are the difference observed between conditions tested statistically significant?

5) In the discussion is reported:...."Interestingly, while SARS-CoV-2 SP only slightly altered the glycolytic shift, it did not appear to reverse the RCC-induced metabolic reprogramming[33]. Although SARS-CoV-2 has been implicated in causing metabolic disturbances in various tissues, including the kidneys, the results of this study suggest that the metabolic reprogramming driven by RCC may overshadow the viral impact on cellular metabolism[34–406 36]."

The inference seems too strong to me, because in the experimental conditions only exposure to the Spike of SARS-CoV-2 is tested. This is a very different condition from the actual viral infection. This should also be reported as a limitation of the present study.

Author Response

Reply: The authors are much appreciated for the constructive appraisal of our manuscript. We have taken in considerations all comments provided and have improved the manuscript accordingly. Point-by-point replies to each comment are provided below.

Comments and Suggestions for Authors

In their work, Dr. Somova and collaborators present a study on the combined effects of exposure to the Spike protein of SARS-CoV-2 (delta variant) in combination with the presence of metastatic renal cell carcinoma cells on the epithelial cells of the renal proximal tubule. The work investigates the effects at the metabolic, immunomodulatory and trascriptional levels.

Although the work exploits a very fine microphysiological system for the study of the renal epithelium, some analyzes are described in an unclear and somewhat too approximate manner.

I have some major concerns about the analyses presented:

1) In the presentation of the data regarding the expression of the ACE2 receptor and the internalization of the Spike protein, the different internalization capacity of the Spike protein between RPTEC and RCC cells is described, but in figure 2 I cannot find the data presented for the RCC spheroids. The lack of internalization of SP in RCC cell is also reported during discussion of the data, but is not shown in the Result section.

RE: In our initial submission the data referring to the lack of Spike protein internalization in RCC spheroids was not included as noted. We have changed figure 2 to representative images of these results (panels G, H, I).

2) Why is gene set enrichment analysis not reported for 3D versus CC? The immunomodulatory effect of CC+SP is described, but what about exposure to CC alone?

Related to these analyses, the gene names in the heatmaps of Figure 4C are not clearly readable.

RE: The gene set enrichment was not reported for the 3D versus CC conditions because no substantial differential gene expression was identified, and no pathways identified as affected using gene set enrichment analysis. In figure 3A the DEG pool of the co-culture condition (CC) and co-culture + spike protein (CC_SP) are represented in the Venn diagram. Further interpretation of the CC vs CC_SP conditions showing the pathways associated the identified DEG is presented in Figure 4A, where the CC was used as the reference conditions for the analysis. Figure 4C has been amended so that the gene names in the heatmaps are legible. The top 40 DEG from the Antigen and T-cell pathways were selected.

3) Why was the level of cytokine secretion not measured only in RCC (-/+ SP) cells? Perhaps this control could help define additional effects. It appears that RCC co-culture always has a greater effect than SARS-CoV2 SP exposure dependence.

RE: Our study sis mainly focused on the effects of the spike protein on healthy renal tubules under the influence of RCC. The exposure to RCC alone was not tested. We concur that this is a valuable control and will be considered in the experimental design of the follow up study. As mentioned, in a previous study we analyzed the expression of TNFa in co-culture and RCC alone, it was determined that our RCC model does not express TNFa (secretion and at the gene level) but induces the expression in the healthy renal tubules.

4)In the metabolic adaptation analysis, in figure 6A-B, the metabolic rate are reported for RPTEC (A) and RCC (B), respectively. Is not clear why Caki-1 (RCC cells) are co-cultured with RCC. Which condition is reported in Figure 6B?. Please clarify better, because is not properly clear. Why tipical Seahorse profiles are not reported? Are the difference observed between conditions tested statistically significant?

RE: The conditions analyzed using the Agilent Seahorse Mito stress test kit are the same presented in preceding figures, and exemplified in figure 1 K-L. The activity or either renal tubules or RCC spheroids was measured following the incubation in the presence or absence of spike protein. The data presented in figure 6 refers to only the proximal tubules (6A) and only the RCC spheroids (6B). We have now clarified the data presentation in the caption of figure 6. In Appendix D we have now the ECAR OCR and PER profiles obtained from the Seahorse experiments for the conditions tested. In the discussion we also comment on the fact that the metabolic data obtained was substantially variable and only 2 biological replicates were used, precluding statistical analysis and informing only about the trends observed in the metabolic shifts.

5) In the discussion is reported:...."Interestingly, while SARS-CoV-2 SP only slightly altered the glycolytic shift, it did not appear to reverse the RCC-induced metabolic reprogramming[33]. Although SARS-CoV-2 has been implicated in causing metabolic disturbances in various tissues, including the kidneys, the results of this study suggest that the metabolic reprogramming driven by RCC may overshadow the viral impact on cellular metabolism[34–406 36]."

The inference seems too strong to me, because in the experimental conditions only exposure to the Spike of SARS-CoV-2 is tested. This is a very different condition from the actual viral infection. This should also be reported as a limitation of the present study.

RE: We do agree that the effects observed are dictated by the single exposure to Spike protein and are not a complete representation of conditions under a viral infection. We now acknowledge this fact in the manuscript discussion and highlight the limitation of the study and how our model is better equipped to infer the activity of spike protein rather than the viral infection per se. 

Reviewer 2 Report

Comments and Suggestions for Authors

I review many papers every year, and it is rare that I find one that is as elegant as this one.  The nature of the in vitro system that the authors have developed is truly novel and is extremely well suited for assessing underlying pathological processes as clearly as this.  The key methodological aspect here is the use of the Humimic chip, which allows controlled perfusion ratesbased on precise pumps. The authors have paired this with 3D-printed chambers that allow for reconstruction of a tubule and individual perfusion of these “tubules”.  RCC spheroids can be added to the system to reproduce a tumor microenvironment and to examine factors secreted from the RCCs in a hydraulic circuit in series with the renal tubular epithelial cells. This system allows the authors to carry out a number of procedures in this very controlled way, including the examination of changes in gene expression in the tubular cells, examination of secretion of secreted factors (e.g. various interleukins, TNF, etc.).  There are so many interesting things that can be done with a system like this.  To test the fundamental hypothesis of this particular study, the authors examined the role of the spike protein (SP) of the SARS-CoV-2 virus (SP).  The justification for this was nicely outline in the Introduction, but in brief, SARS-CoV2 infection produces high rates of renal morbidity, especially acute kidney injuries affecting the proximal tubule.  

The authors were able to identify 100+ genes that appeared to be specifically upregulated by the combined presence of SP and a RCC spheroid in the system.   Most of these are genes are already implicated in immune function and cancer biology.  The authors obtained data strongly suggesting that the effect of SP is mediated on the proximal tubule cells rather than on the tumor spheroids.  An aspect that I found particularly interesting was the ability to identify changes in secretions of certain cytokines, in particular changes in interleukins 1, 5, 6, and 8, as well as TNF.  The pattern of changes in cytokine secretion was complex.  It will take some time in subsequent studies to unravel the significance of the changes, i.e. they weren't all increased.  The power of this system though is that it allowed the authors to identify a complex pattern of this sort, which would have been technically impossible in an in vivo system -- certainly they would not have been able to readily examine as many factors simultaneously.  This is an important strength of the study, as changes in a tumor microenvironment are just that -- they are an altered milieu rather than alterations in just one thing.  Granted this is just the start of understanding the consequences of these changes, but it is an essential and first step.  Finally, the authors were able to show that there were specific metabolic changes associated with the combined presence of tumor spheroids and SP on the reconstructed tubules, notably (and not surprisingly) including a marked increase in glycolytic as opposed to oxidative metabolism.

It took me some time to go through the MS, and in some measure this is because I am not an expert on renal cell carcinoma or SARS-CoV-19.  But the experimental designs are sound, the data are presented clearly and support the conclusions, and the writing and illustrations are quite clear.

This is one of those extremely rare cases where I really can't make any suggestions for improvement.    My hats are off to the authors for what they did here.

Author Response

Reply: We appreciated very much the positive revision of our manuscript and are grateful for the commendation of our work. We´d like to add that we continue to improve our RCC model so it can be even more representative of kidney cancer pathophysiology. The next iteration of our model, being used in our follow up experiments will include an immune component, in the form of circulating immune cells that can interact with the RCC spheroids and renal tubules, in order to capture any immune activation effects elicited.

Reviewer 3 Report

Comments and Suggestions for Authors

The manuscript “SARS-CoV-2 Spike Protein Amplifies the Immunogenicity of Healthy Renal Epithelium in the Presence of Renal Cell Carcinoma” by Somova, et.al. is dedicated to the establishment and characterization of the co-culture system of renal carcinoma cells and non-malignant kidney cells. Authors also use the established system to characterize the effect of SARS-CoV-2 Spike protein on the abovementioned system. The manuscript is an effort to develop reliable, reproducible and easy to use system to study the effect of COVID-19 on different organs and tissues when several pathologies developed in one patient. The microfluidics is a perfect solution for such type of research as it provides robust system to culture different cell types and collect samples at different stages of the cultivation.  

Somova and co-authors utilized different methods to describe the co-culture itself, RNA-seq, ELISA of cytokines, microscopy and metabolomics among them. The data analysis and statistical analysis were performed at appropriate level and allow to evaluate the whole picture. Since the aim of the study was to characterize the system and sample the effect of coronavirus spike protein on the cells, the study is purely descriptive. Authors do not propose any mechanism of S-protein effect on the carcinoma or kidney cells, they merely use it as proof of concept to test the system. 

The manuscript is important because it expands our understanding of the RABV G protein influence on the host neurons. Liu and coauthors developed a series of recombinant proteins from different strains of the infection that were shown different levels of disease severity. Authors utilized the advanced techniques, such as proteomics, to explore possible protein candidates involved to pathogenicity. 

The major issues:

-              Some figures with gene names are not readable (e.g., Figure4C).

-              Authors need to define which abbreviations they want to use. The SP is rather non-canonical way to name the protein, spike or S protein (see in https://doi.org/10.1038/s41401-020-0485-4). There is an established way to name viral proteins. 

-              Also, authors should follow the common in the field way to name the SARS-CoV-2, unlike other ways they use (lines 149, 205, 266, 304, 355, etc.). Generally, following uniform nomenclature is highly recommended.

-              The English needs corrections. Also, Introduction is not comprehensive. Authors need to improve it and make more fluent.

The minor issues:

Line 117 – CO2.

Line 197 – make everything uniform, data from capital. 

Figure 1, 6C – whichever color code you use, it should be explained in the legend.

Figure 4A – sign the y-axis.

To sum up, the system presented by authors is a promising prototype to study SARS-CoV-2 effect on diverse cell types. The effect of spike protein that sheds during COVID-19 is highly understudied exactly because it is hard to access in the regular cell cultures or in animal models. Therefore, I believe the present research will be of great interest to the scientific community.

Author Response

Reply: The authors are much appreciated for the constructive appraisal of our manuscript. We have taken in considerations all comments provided and have improved the manuscript accordingly. Point-by-point replies to each comment are provided below. 

Comments and Suggestions for Authors

The manuscript “SARS-CoV-2 Spike Protein Amplifies the Immunogenicity of Healthy Renal Epithelium in the Presence of Renal Cell Carcinoma” by Somova, et.al. is dedicated to the establishment and characterization of the co-culture system of renal carcinoma cells and non-malignant kidney cells. Authors also use the established system to characterize the effect of SARS-CoV-2 Spike protein on the abovementioned system. The manuscript is an effort to develop reliable, reproducible and easy to use system to study the effect of COVID-19 on different organs and tissues when several pathologies developed in one patient. The microfluidics is a perfect solution for such type of research as it provides robust system to culture different cell types and collect samples at different stages of the cultivation.  

Somova and co-authors utilized different methods to describe the co-culture itself, RNA-seq, ELISA of cytokines, microscopy and metabolomics among them. The data analysis and statistical analysis were performed at appropriate level and allow to evaluate the whole picture. Since the aim of the study was to characterize the system and sample the effect of coronavirus spike protein on the cells, the study is purely descriptive. Authors do not propose any mechanism of S-protein effect on the carcinoma or kidney cells, they merely use it as proof of concept to test the system. 

The manuscript is important because it expands our understanding of the RABV G protein influence on the host neurons. Liu and coauthors developed a series of recombinant proteins from different strains of the infection that were shown different levels of disease severity. Authors utilized the advanced techniques, such as proteomics, to explore possible protein candidates involved to pathogenicity. 

The major issues:

-              Some figures with gene names are not readable (e.g., Figure4C). 

RE: Figure 4C has been amended so that the gene names in the heatmaps are legible. The top 40 DEG from the Antigen and T-cell pathways were selected.

-              Authors need to define which abbreviations they want to use. The SP is rather non-canonical way to name the protein, spike or S protein (see in https://doi.org/10.1038/s41401-020-0485-4). There is an established way to name viral proteins. 

RE: The abbreviations have now been altered to comply with naming conventions

-              Also, authors should follow the common in the field way to name the SARS-CoV-2, unlike other ways they use (lines 149, 205, 266, 304, 355, etc.). Generally, following uniform nomenclature is highly recommended.

RE: The abbreviations have now been altered to comply with naming conventions and made uniform across the manuscript

-              The English needs corrections. Also, Introduction is not comprehensive. Authors need to improve it and make more fluent.

RE: We have revised the introduction and overall manuscript to improve readability and remove any grammatical miss constructions.

The minor issues:

Line 117 – CO2.

Line 197 – make everything uniform, data from capital. 

Figure 1, 6C – whichever color code you use, it should be explained in the legend.

Figure 4A – sign the y-axis.

RE: All the minor issues have been addressed

To sum up, the system presented by authors is a promising prototype to study SARS-CoV-2 effect on diverse cell types. The effect of spike protein that sheds during COVID-19 is highly understudied exactly because it is hard to access in the regular cell cultures or in animal models. Therefore, I believe the present research will be of great interest to the scientific community.

Round 2

Reviewer 1 Report

Comments and Suggestions for Authors

The authors have more or less answered all the questions raised. The tone of the work was rightly adapted to the data presented.

The figures are definitely more readable.

The axis of plot B must be placed in appendix D, aligning it with the others, otherwise it is unreadable.

Observing the ECAR, OCR and PER profiles (Appendix D, D-F), an evident effect of RPTEC co-colture condition in RCC spheroids is observed. The same effect seem to revert with the addiction of the Spike protein. Are these data discussed? In the discussion is reported that: "On the other hand, RCC activity was consistently more stable....". Is correct?

Numerous typos, missing spaces and punctuation errors still remain in the text. Please check.

Pay particular attention to the punctuation of the figure legends (e.g. 1,2 and 5).

Comments on the Quality of English Language

The quality of English seems to have improved. However, I recommend re-reading it to a native English speaker. 

There are still many typos.

Author Response

Reply: We once again appreciated the comments provided and please find bellow a point by point reply to the issues raised.

The authors have more or less answered all the questions raised. The tone of the work was rightly adapted to the data presented.

The figures are definitely more readable.

The axis of plot B must be placed in appendix D, aligning it with the others, otherwise it is unreadable.

RE: The plot in appendix has been correct so that the x-axis of panel is legible.

Observing the ECAR, OCR and PER profiles (Appendix D, D-F), an evident effect of RPTEC co-colture condition in RCC spheroids is observed. The same effect seem to revert with the addiction of the Spike protein. Are these data discussed? In the discussion is reported that: "On the other hand, RCC activity was consistently more stable....". Is correct?

RE: These observations have been reiterated in the result section (lines 337-339) and have been considered in our discussion. We have also clarified in the discussion that what we mean by stating that RCC activity is more stable, is the fact that the data obtained for the RCC during our assays was less variable relative to the results obtained from the renal tubules.

Numerous typos, missing spaces and punctuation errors still remain in the text. Please check.

Pay particular attention to the punctuation of the figure legends (e.g. 1,2 and 5).

RE: We have once again proof-read the manuscript and amended any typos and textual miss constructions. Figures legends have been corrected.

Comments on the Quality of English Language

The quality of English seems to have improved. However, I recommend re-reading it to a native English speaker. 

There are still many typos.

RE: The manuscript has been also proof-read by an English native speaker to ensure a smooth readability.